# Gene Losses and Plastome Degradation in the Hemiparasitic Species *Plicosepalus acaciae* and *Plicosepalus curviflorus*: Comparative Analyses and Phylogenetic Relationships among Santalales Members

**DOI:** 10.3390/plants11141869

**Published:** 2022-07-18

**Authors:** Widad AL-Juhani, Noha T. Al Thagafi, Rahmah N. Al-Qthanin

**Affiliations:** 1Department of Biology, Faculty of Applied Science, Umm Al-Qura University, Makkah 24381, Saudi Arabia; 2Department of Biology, Faculty of Science, Taif University, Taif 21974, Saudi Arabia; noha.t@tu.edu.sa; 3Department of Biology, College of Sciences, King Khalid University, Abha 61421, Saudi Arabia; rngerse@kku.edu.sa; 4Prince Sultan Bin-Abdul-Aziz Center for Environment and Tourism Studies and Researches, King Khalid University, Abha 61421, Saudi Arabia

**Keywords:** plastome, *Plicosepalus acaciae*, *Plicosepalus curviflorus*, loranthaceae, mistletoe, phylogenetic relationship, plastome structure, comparative analysis

## Abstract

The *Plicosepalus* genus includes hemiparasitic mistletoe and belongs to the Loranthaceae family, and it has several medicinal uses. In the present study, we sequenced the complete plastomes of two species, *Plicosepalus acaciae* and *Plicosepalus curviflorus*, and compared them with the plastomes of photosynthetic species (hemiparasites) and nonphotosynthetic species (holoparasites) in the order Santalales. The complete chloroplast genomes of *P*. *acaciae* and *P*. *curviflorus* are circular molecules with lengths of 120,181 bp and 121,086 bp, respectively, containing 106 and 108 genes and 63 protein-coding genes, including 25 tRNA and 4 rRNA genes for each species. We observed a reduction in the genome size of *P*. *acaciae* and *P*. *curviflorus* and the loss of certain genes, although this reduction was less than that in the hemiparasite and holoparasitic cp genomes of the Santalales order. Phylogenetic analysis supported the taxonomic state of *P*. *acaciae* and *P*. *curviflorus* as members of the family Loranthaceae and tribe Lorantheae; however, the taxonomic status of certain tribes of Loranthaceae must be reconsidered and the species that belong to it must be verified. Furthermore, available chloroplast genome data of parasitic plants could help to strengthen efforts in weed management and encourage biotechnology research to improve host resistance.

## 1. Introduction

Parasitic plants completely or partially lose the ability to photosynthesize, and they absorb water and nutrients from the host via the haustorium. Depending on the degree of loss of photosynthetic ability, parasitic plants are divided into photosynthetic parasites (hemiparasites) and nonphotosynthetic parasites (holoparasites) [1]. All hemiparasitic species are capable of photosynthesis (at different levels), whereas holoparasites entirely lose their photosynthetic ability and obtain all nutrients from their hosts [2]. The most parasitic plant species are included in the family Orobanchaceae and the order Santalales.

The family Loranthaceae is the largest in the sandalwood order Santalales, and it includes 76 genera and over 1000 species [3,4]. Loranthaceae are mainly distributed in tropical and subtropical regions of the Americas, Africa, Asia and Australia, with a few species extending to the temperate zones in Europe and East Asia [3,5]. The origin of the Loranthaceae family has been traced back to the Australasian continent in the Late Cretaceous, and it was likely dispersed by birds from Australasia to Asia. African and European species migrated from Asia after the middle Oligocene, and birds also played an important role in dispersing Loranthaceae from Africa to Madagascar [3]. Loranthaceae mainly include aerial parasitic plants, although they also contain three monotypic genera that are root parasites and considered relicts [5]. Mistletoe is an aerial parasitic plant that depends on pollination and seed dispersal by birds and other pollinators, and it provides a habitat for these organisms; moreover, mistletoe determines the natural structure of the surrounding plant communities [6].

In terms of taxonomic state, initial studies on Loranthaceae included most mistletoes in the genus *Loranthus* or in a section of *Loranthus* [7,8] presented genera with new names, some of which are still used at present. However, the interspecific relationships within genera remain unclear [4].

The Loranthaceae family was divided into tribes and subtribes [5,9] according to morphological data, chromosome numbers, and molecular sequences of the “nuclear gene rDNA and chloroplast genes *rbcl*, *matk*, and *trnl-F*”. The tribe Lorantheae has been characterized by a chromosome number x = 9, the tribe Psittacantheae has been characterized by x = 8, while the other two tribes (Elytrantheae and Nuytsieae) have been characterized by x = 12 [5].

*P*. *acaciae* (Zucc.) Wiens and Polhill, and *P*. *curviflorus* (Benth. ex Oliv.) Tiegh, are hemiparasite species (mistletoe) belonging to Loranthaceae. These plants are woody parasites with brownish slender branchlets, evergreen, rigidly coriaceous, oblong to elliptic-oblong, glabrous leaves, solitary or clustered umbels of scarlet flowers, and berry fruits, and they are widespread parasites of *Acacia* [10]. *P*. *acaciae* is distributed in Syria to northeast tropical Africa and the southern Arabian Peninsula. *P*. *curviflorus* is distributed in southeast Egypt to east central and east tropical Africa. *Plicosepalus* is traditionally used as a treatment for diabetes and cancer and as an antioxidant, antimicrobial, anti-inflammatory agent, and it is also used to increase lactation in cattle [11,12].

According to the parsimony, likelihood and Bayesian inference trees presented by [5], *Plicosepalus sagittiflorus* belongs to subtribe Loranthinae (Clade J) and Asian and African taxa (x = 9). Furthermore, this study reported 12 species belonging to the genus *Plicosepalus*, which is the same number currently mentioned in Plants of the World Online [13].

Liu et al. conducted taxonomic analyses of Loranthaceae based on floral, inflorescence morphology, phylogenetic and biogeographical analyses using both nuclear and chloroplast DNA regions, and they included 62 of the 76 genera [3]. This study classified the species belonging to family Loranthaceae within five tribes (Psittacantheae, Nuytsieae, Elytrantheae, Gaiadendreae and Lorantheae), and species belonging to the tribe Lorantheae were distributed within seven subtribes (Tapinanthinae, Scurrulinae, Emelianthinae, Dendrophthoinae, Amyeminae, Loranthinae and Ileostylinae). The species *Plicosepalus sagittiflorus* and *P*. *curviflorus* are representative of the genus *Plicosepalus*, which appears in the phylogenetic tree within the subtribe Tapinanthinae and belongs to the Africa–Madagascar section (including regions of the coastal area of the Arabian Peninsula and Sub-Saharan Africa and Madagascar). However, the African subtribe Tapinanthinae was not monophyletic but rather was nested within the Emelianthinae subtribe. Therefore, further studies are recommended to resolve the phylogenetic relationship between Tapinanthinae and Emelianthinae and test the monophyly of the African genera.

The chloroplast is an organelle in plant cells that contains its own genome, the plastome, and it has essential roles in photosynthesis [14]. During the evolutionary history of plant families, plastomes have been subjected to strong selective pressures [15]. Thus, chloroplast genomes include useful phylogenetic information that has been used to study evolutionary relationships at different taxonomic levels and resolve difficult problems in plant phylogenetics [3,16]. Investigating the genomes of hemiparasitic plants is important for understanding the changes in the chloroplast genome of a plant when it shifts from an autotrophic to a parasitic state. Moreover, most parasitic plants are medicinally important, and their reliable identification is important for avoiding damage caused by the misidentification of herbal medicinal plants. This study aimed to assemble and compare the complete plastomes of *P*. *acaciae* and *P*. *curviflorus*; assess the systematic relationships within the family and tribes of Loranthaceae; and compare the new plastomes with representatives of hemiparasites and holoparasites to investigate the evolution of plastomes associated with parasitism.

## 2. Results

### 2.1. Characteristics of the Chloroplast Genome

The complete chloroplast genomes of *P*. *acaciae* and *P*. *curviflorus* are shown in Figure 1, and they are circular molecules with lengths of 120, 181 bp and 121,086 bp, respectively. The chloroplast genome presents a typical four-region structure that consists of a large single copy (LSC), a small single copy (SSC) and two inverted repeats (IRa and IRb). The LSC and SSC regions in *P*. *acaciae* and *P*. *curviflorus* are 69,497 bp and 69,947 bp long and 6038 bp and 6187 bp long, respectively, while IRa and IRb are 22,323 bp and 22,476 bp each in *P*. *acaciae* and *P*. *curviflorus* (Table 1). The lengths of the coding regions are 58,089 and 64,539 bp, and they represent 48.33% and 53.30% of the entire genome in *P*. *acaciae* and *P*. *curviflorus*, respectively, while the noncoding region lengths are 62,092 and 56,547 bp (51.67% and 46.70%), respectively. The percentage of AT in *P*. *acaciae* and *P*. *curviflorus* in the entire genome is 63.43% and 63.23%, respectively, whereas the percentage of GC is 36.6% and 36.8%, respectively. The genomic structure of *P*. *acaciae* and *P*. *curviflorus* consists of A = (31.50% and 31.34%), T(U) = (31.93% and 31.89%), C = (18.59% and18.65%) and G = (17.97% and18.12%), respectively, as shown in Table 1.

Figure 1 and Appendix A show the results obtained from gene annotation of the chloroplast genomes of *P*. *acaciae* and *P*. *curviflorus*: 106 and 108 genes were obtained, respectively, which included 92 of unique genes and 12 and 14 genes that were duplicated in the IR region. The cp contains 63 protein-coding genes, including 25 tRNA and 4 rRNA genes for each species. Most of the protein-coding genes start with a methionine codon (AUG).

Introns play a significant role in gene expression regulation [17,18]. Table 2 and Table 3 illustrate that 10 of the 106 and 108 genes in *P*. *acaciae* and *P*. *curviflorus* contain introns, and they include 8 protein-coding genes and 2 tRNA genes. The *ycf3* and *clpP* genes have two introns, while the remaining genes have only one intron. The seven introns are included in the LSC, and the rest are specifically located within the IRa and IRb regions.

### 2.2. Relative Synonymous Codon Usage (RSCU)

Codon usage bias plays an important role in chloroplast genome evolution and occurs as a result of natural selection and mutations [19,20]. The nucleotides of protein-coding and tRNA genes (58,089 bp and 64,539 bp, respectively) in *P*. *acaciae* and *P*. *curviflorus* were used to determine the codon usage bias of the plastome. As shown in Appendix A, these genes are encoded by 19,363 and 21,513 codons in *P*. *acaciae* and *P*. *curviflorus*, respectively. As shown in Figure 2, the amino acid leucine was the most frequent amino acid (10.097% and 10.052%, respectively) in *P*. *acaciae* and *P*. *curviflorus*, cysteine was the least frequent amino acid (1.097%) in *P*. *acaciae*, and tryptophan (1.181) was the least frequent amino acid in *P*. *curviflorus*.

The RSCU values in *P*. *acaciae* in Appendix A show that 27 codons are >1, while 32 codons are <1. The data show that tryptophan, glycine, valine, and methionine with no codon usage bias have an RSCU value of 1. The RSCU values in *P*. *curviflorus* in Appendix A show that 26 codons are >1 and 32 codons are <1. The data show that tryptophan, glycine, and methionine with no codon usage bias have an RSCU value of 1.

### 2.3. RNA Editing Sites

RNA editing sites include the processes of inserting, deleting or modifying nucleotides, which lead to changes in the DNA coding sequence during RNA transcription processes [21], which in turn allows for the creation of different protein transcripts [22]. The PREP suite was used to predict the RNA editing sites in the *P*. *acaciae* and *P*. *curviflorus* plastomes, and the first codon position of the first nucleotide was used in the analysis. The RNA editing sites are presented in Appendix A. Overall, there are 33 and 40 editing sites in the genomes of *P*. *acaciae* and *P*. *curviflorus* distributed among 15 and 16 protein-coding genes. The results show that the highest number of editing sites in *P*. *curviflorus* are *the rpoB* and *rpoC2* genes (7 and 6 sites, respectively), while the highest number of editing sites in *P*. *acaciae* are the *rpoB* and *matK* genes (6 and 5 sites). Most of the codon position exchanges involve the amino acids serine (S) and leucine (L) (S to L). The results of RNA editing show that certain genes do not possess a predictable site in the first nucleotide of the first codon, namely, *rpl23*, *accD*, *atpB*, *clpP*, *petG*, *petL*, *psaB*, *psaI*, *psbE*, *psbF*, and *rps14 in P*. *acaciae* and *accD*, *atpB*, *clpP*, *ndhA*, *ndhB*, *ndhD*, *ndhF*, *ndhG*, *petD*, *petG*, *petL*, *psaB*, *psaI*, *psbE*, *psbF*, *psbL*, *rpl2*, *rpl23*, and *rps16* in *P*. *curviflorus*.

### 2.4. Repeat Analysis

#### 2.4.1. Long Repeats

Figure 3 shows that there are four types of repeats in the *P*. *acaciae* and *P*. *curviflorus* cp genomes: palindromic (16 and 11), forward (12 and 13), reverse (16 and 19), and complementary (4 and 5). We compared the frequency of repeats (palindromic, forward, reverse, and complementary) in the cp genomes of *P*. *acaciae*, *P*. *curviflorus* and six species of Loranthaceae (*Scurrula chingii* (W.C.Cheng) H.S.Kiu, *Taxillus chinensis* (DC.) Danser, *Loranthus europaeus* Jacq., *Dendrophthoe pentandra* (L.) Miq., *Nuytsia floribunda* (Labill.) R.Br. ex G.Don. *Elytranthe albida* (Blume) Blume.) (Figure 3). *T*. *chinensis* had the highest frequency of palindromic repeats (24), *N*. *floribunda* had the highest frequency of forward repeats (16), and *P*. *curviflorus* had the highest frequency of reverse repeats (19). Complement repeats were the least common type of repeat in the genome, with *P*. *curviflorus* and *E*. *albida* having five repeats and *N*. *floribunda* presenting no complementary repeats (Figure 3).

#### 2.4.2. Simple Sequence Repeats

A total of 164 and 155 SSRs are present in the plastid genome of *P*. *acacia* and *P*. *curviflorus*, as shown in Table 4, poly T (42.07 and 52.24%) and A (40.85 and 46.27%), and poly C (1.22 and 1.49%) and G (0.61 and 0.00%) repeats. The results obtained from the analysis of microsatellite frequency in the genome of six species from the Loranthaceae family are presented in Table 4. The di-repeat AT/AT is found in the genome of all species, while the dinucleotide AG/CT is found in five species but is absent in three species: *S*. *chingii*, *T*. *chinensis* and *E*. *albida*. Furthermore, there were 3 trinucleotide repeats (AAG/CTT, AAT/ATT and ATC/ATG), 11 tetra repeats (AAAG/CTTT, AAAT/ATTT, ACAG/CTGT, AGGG/CCCT, AAGT/ACTT, AATT/AATT, AATC/ATTG, AAAC/GTTT, AACC/GGTT, AGAT/ATCT, and AATG/ATTC) and 2 penta repeats (AAAAT/ATTTT and AATAT/ATATT) (Table 3).

The frequency of SSRs among the cp genomes of the eight species was also compared (Figure 4), and the results showed that mononucleotides occurred more frequently across all genomes. *N*. *floribunda* had the highest number of dinucleotides with 13 repeats, *P*. *acacia* has the highest number of trinucleotides with 6 repeats, and L. europaeus had the highest number of tetranucleotides with 12 repeats and pentanucleotides with 4 repeats. The majority of SSRs in the cp genome *of P*. *acacia* and *P*. *curviflorus* were monorepeats (84.76 and 86.45%) (Figure 5).

### 2.5. Sequence Divergence

To investigate the degree of sequence divergence, the program mVISTA was used to align the complete chloroplast genomic sequences of *P*. *acaciae* and *P*. *curviflorus* with the six Loranthaceae chloroplast genomes available in GenBank: *S*. *chingii*, *T*. *chinensis*, *L*. *europaeus*, *D*. *pentandra*, *N*. *floribunda* and *E*. *albida*. The annotation of *P*. *acaciae* was used as a reference. The protein-coding genes were more conserved than the noncoding regions (Figure 6). The noncoding regions presented high divergence in the following genes: *trnH-GUG-matK*, *matK-trnQ-UUG*, *atpA-atpH*, *rpoC2-rpoB*, *rpoB-trnC-GCA*, *trnT-GGU-psbD*, *psbC-trnS-UGA*, *pspA-ycf3*, *ycf3-trnS-UGU*, *atpB-rbcL*, *rbcL-rpl36*, *rpl36-pspI*, *pspI-ycf4*, *ycf2-trnL-CAA*, *trnL-CAA-rps7*, *ycf15-trnV-GAC*, *rps12-ycf15*, *ycf15-trnV-GAC*, *trnV-GAC-rrn23S*, *trnR-ACG-trnN-GUU*, *trnN-GUU-trnR-AGG*, *petA-psbJ*, *pspE-petL*, *rpl20-rps12*, *rps12-clpP*, *clpP-psbB*, *psbN-petB*, *petB-petD*, *rpl14-rps3*, *trnA-UGC-rrn16S*, *trnV-GAC-ycf15*, *ycf15-rps12*, *rps12-rps7*, *rps7-trnL-CAA*, *trnL-CAA-rpl23* and *rpl23-trnH-GUG*. However, the protein-coding genes showed divergence in fewer regions: *matK*, *rpoC2*, *accD-rpl36*, *ycf2* and *ycf1*.

### 2.6. Boundary between LSC/SSC and IRs

The difference in the lengths of the LSC, SSC, and IR regions between the species was compared (Figure 7), and a significant reduction in the size of the SSC region was observed, with the shortest SSC (5250 bp) in *E*. *albida* and the longest SSC in *P*. *curviflorus* (6187 bp). *E*. *albida* had the largest LSC region at 72,966 bp, while *D*. *pentandra* had the shortest LSC at 69,368 bp. Great variation in the length of the IR regions was observed among the species, with the root hemiparasite *N*. *floribunda* presenting the largest IR region at 26,801 bp and *D*. *pentandra* presenting the shortest IR region at 20,118 bp.

The four genes *rp12*, *trnL*, *ycf1*, and *trnH*, which were located at the junction of inverted repeats and single copy regions, showed variations in the location and number of base pairs, with the IRa-SSC and IRb border presenting the greatest variations. The two genes *ycf1* and *trnL* were found in the IR/SSC border. The *ycf1* gene crossed the SSC/IRa border in *D*. *pentandra* (873 bp), *E*. *albida* (2310 bp), *L*. *europaeus* (963 bp), *N*. *floribunda* (2982 bp) and *S*. *chingii* (945 bp). The SSC/IRa border was distinct in *P*. *acaciae* and *P*. *parviflorus* along the *rps15* gene. The *ycf1* gene was in the IRb region of *D*. *pentandra* (872 bp), *E*. *albida* (2309 bp) and *S*. *chingii* (945 bp), while the extended *ycf1* gene was observed in the SSC region of *P*. *acaciae* and *P*. *curviflorus* (4070 bp and 4112 bp, respectively).

The *trnL* gene was located in the SSC region in *D*. *pentandra*, *E*. *albida* and *S*. *chingii*, and it was located across the SSC/IRb in *T*. *chinensis* (15 bp) and *L*. *europaeus* (45 bp). The root parasite *N*. *floribunda* differed from the other species, with the *rpI32* gene located across the SSC/IRb (26 bp). The *rp12* gene extended across the LSC/IRb in all species: *P*. *acaciae* (218 bp), *P*. *curviflorus* (228 bp), *D*. *pentandra* (231 bp), *E*. *albida* (300 bp), *L*. *europaeus* (215 bp), *N*. *floribunda* (225 bp), *S*. *chingii* (249 bp) and *T*. *chinensis* (219 bp). There was a duplication in the rp12 gene that was present as well on the LSC/IRa border in *N*. *floribunda*.

Variations were observed in the location of the *trnH* gene in the IRa/LSC border as well, and it was in the LSC region in *S*. *chingii* and *E*. *albida*, across the IRa/LSC border in *D*. *pentandra*, and extended across the IRa region away from the IRa/LSC border (1 bp) in *P*. *acaciae*, *P*. *curviflorus*, *N*. *floribunda* and *L*. *europaeus*.

### 2.7. Divergence of Protein-Coding Gene Sequences

The value of synonymous (Ks) and nonsynonymous (Ka) substitutions and the Ka/Ks ratio were calculated among the 59 protein-coding genes that represent the common genes in the chloroplast genomes of *P*. *acaciae*, *P*. *curviflorus* and six Loranthaceae species *S*. *chingii*, *T*. *chinensis*, *L*. *europaeus*, *D*. *pentandra*, *N*. *floribunda* and *E*. *albida*. The Ka/Ks value usually used for evaluating sequences variations in different species or taxonomical species with unknown evolutionary status, and to detect substitution, selection and beneficial mutation genes under selective pressure [23]. As shown in Figure 8, for *Plicosepalus acacia* vs. *Plicosepalus curviflorus, Scurrula chingii, Taxillus chinensis, Loranthus europaeus, Dendrophthoe pentandra, Nuytsia floribunda* and *Elytranthe*, the Ka/Ks ratio was >1 in five genes—petB, psbM, ycf1, rpl23 and atpI; while for *Plicosepalus curviflorus* vs. *Plicosepalus acacia, Scurrula chingii, Taxillus chinensis, Loranthus europaeus, Dendrophthoe pentandra, Nuytsia floribunda* and *Elytranthe* in Appendix A,the Ka/Ks ratio was >1 in three genes—petB, ycf1 and accD. However, all the Ks values were <1 in all of the genes (Figure 8 and Appendix A).

### 2.8. Heatmap

The heatmap was created to investigate the evolution of the plastome associated with parasitism in the aerial hemiparasites investigated in this study, i.e., *P*. *acacia* and *P*. *curviflorus*, and they were compared to representatives of the Loranthaceae family: aerial hemiparasitic *S*. *chingii*, *T*. *chinensis*, *L*. *europaeus*, *D*. *pentandra* and *E*. *albida*. *N*. *floribunda* represents a root hemiparasite in Loranthaceae. In addition to the species *Viscum album* as an aerial hemiparasitic (Viscaceae), *Schoepfia jasminodora* is a root hemiparasite (Santalaceae). *Erythropalum scandens* (Santalales; Erythropalaceae) is an example of an autotrophic plant. In contrast, *Epifagus virginiana* was used in the heatmap as a representative of an obligate parasite or holoparasite (Orobanchaceae), (GenBank accession numbers, names are available in Table 5).

As shown in Figure 9, common losses were observed in some protein-coding and tRNA genes as follows: the gene group *ndh* (A, B, C, D, F, H, I, J and K) was absent from most species or present as pseudogenes in some cases, such as the *ndhB* gene pseudogene in *E*. *virginiana*, *L*. *europaeus*, and *N*. *floribunda* and *ndhA* gene in *S*. *jasminodora*. The *infA* gene was absent in most species and pseudogenes in *L*. *europaeus*, whereas it was present in *E*. *virginiana*, *N*. *floribunda* and *S*. *jasminodora*.

The *rpl16* gene was present in *E*. *virginiana*, *P*. *acacia*, *P*. *curviflorus*, *D*. *pentandra*, *E*. *albida* and *N*. *floribunda* and *Schoepfia jasminodora*, and the pseudogene in *S*. *chingii*, *T*. *chinensis*, and *L*. *europaeus* was absent from only *V*. *album*. The *rpl32* gene was missing from all comparison species except *N*. *floribunda* and *S*. *jasminodora*. The *rps16* gene was missing from all comparison species except *S*. *jasminodora*. The *rps15* gene was present in only *P*. *acacia*, *P*. *curviflorus* and *S*. *jasminodora*.

Regarding the tRNA genes, *TrnV-UAC* was lost in all comparison species. The *trnG-UCC* gene was absent in all comparison species except *S*. *jasminodora*. The *trnI-GAU* gene was absent in all comparison species except *N*. floribunda. The *trnK-UUU* gene was absent in all comparison species except *T*. *chinensis*, *L*. *europaeus* and *N*. *floribunda*. The *trnA-UGC* gene was present in *P*. *acaciae*, *P*. *curviflorus*, *T*. *chinensis* and *N*. *floribunda* and absent in *E*. *virginiana*, *S*. *chingii*, *D*. *pentandra*, *E*. *albida*, *V*. *album* and *S*. *jasminodora*, while the pseudogene was present in *L*. *europaeus*. The *trnH-GUG* gene was present in *P*. *acaciae*, *P*. *curviflorus*, *S*. *chingii*, *S*. *jasminodora* and *N*. *floribunda*. In contrast, it was absent in *E*. *virginiana*, *T*. *chinensis*, *D*. *pentandra*, *E*. *albida* and *V*. *album*, while the pseudogene was present in *L*. *europaeus*. The gene *trnL-UAG* was present in *P*. *acaciae*, *Plicosepalus curviflorus*, *S*. *chingii*, *T*. *chinensis*, *L*. *europaeus*, *N*. *floribunda* and *S*. *jasminodora*. However, it was absent in *E*. *virginiana*, *D*. *pentandra*, *E*. *albida* and *V*. *album*.

The trnG-GCC gene was present in *P*. *acacia*, *P*. *curviflorus*, *S*. *chingii*, *T*. *chinensis* and *N*. *floribunda*, while the pseudogene was presents in *L*. *europaeus* but absent in *E*. *virginiana*, *D*. *pentandra*, *E*. *albida*, *V*. *album* and *S*. *jasminodora*. The trnI-CAU gene was absent only in *V*. *album* and *E*. *virginiana*. The trnI-UAA gene was present in *P*. *acacia*, *P*. *curviflorus*, *L*. *europaeus*, *N*. *floribunda* and *S*. *jasminodora* but absent in *D*. *pentandra*, *E*. *albida*, *V*. *album*, S. *chingii*, *T*. chinensis and *E*. *virginiana*. The trnP-UGG gene was present in *P*. *acacia*, *P*. *curviflorus*, *S*. *chingii*, T. chinensis, *N*. *floribunda* and *L*. *europaeus*, while it was absent in *E*. *virginiana*, *D*. *pentandra*, *E*. *albida*, *V*. *album* and *S*. *jasminodora*.

### 2.9. Phylogenetic Analysis

In the current study, the phylogenetic relationships were assessed among plastomes of *P*. *acaciae* and *P*. *curviflorus*, and examples of cp genomes of the Santalales order from GenBank included 15 species belonging to the Loranthaceae family, and they represented different tribes (GenBank accession numbers, names are available in Table 5). The cp genomes from three families (Viscaceae, Santalaceae, and Schoepfiaceae) were also included. The *Nicotiana tabacum* chloroplast genome represents an outgroup.

The phylogenetic trees with all nodes having 100% bootstrap support BS (Figure 10). The monophyly of the Loranthaceae family was strongly supported (BS = 100) by the results.

Loranthaceae was divided into three main highly supported clades (BS = 100) in the maximum likelihood (ML) tree: one clade contained species from the Lorantheae tribe, and the other clades consisted of *N*. *floribunda* representing tribe Nuytsieae and *E*. *albida* representing tribe Elytrantheae.

Lorantheae was further divided into two separate clades, and each clade was divided into two subclades representing four different subtribes that were highly supported. The first clade was divided into two subclades, with BS = 100. The first subclade included the subtribe Scurrulinae (BS = 100), which consisted of three species of the genera *Taxillus* (*Taxillus chinensis*, *Taxillus pseudochinensis*, and *Taxillus tsaii*) and *S*. *chingii* a sister, and the second subclade included *Helixanthera parasitica*, representing the subtribe Dendrophthoinae (BS = 100). The second subclade was divided into two branches (BS = 100) consisting of *P*. *acacia* and *P*. *curviflorus* (subtribe Tapinanthinae) and linked to *Moquiniella rubra* (subtribe Emelianthinae) as a sister. The second branch in subclade two included *D*. *pentandra* (subtribe Dendrophthoinae); interestingly, *Macrosolen cochinchinensis*, which belongs to Elytrantheae, was nested with this branch. The second clade of the tribe Lorantheae was divided into two subclades (BS = 100) consisting of four species of the genus *Loranthus* (*Loranthus europaeus*, *Loranthus pseudo-odouratus*, *Loranthus guizhouensis*, and *Loranthus delavayi*) and *Cecarria obtusifolia* as a sister. All species that belonged to this clade represented subtribe (Loranthinae)

## 3. Discussion

This study presents the first chloroplast genome of the species *P*. *acaciae* and *P*. *curviflorus* of the Loranthaceae family. The size, structure, gene content, and organization are usually conserved in the chloroplast genome of angiosperms [15].

Typically, chloroplast genome sizes range between 120 and 170 kilobase pairs (kb) [23]. The length of the cp genome is 120,181 bp and 121,086 bp in *P*. *acaciae* and *P*. *curviflorus*, respectively. The decrease in the size of the genome is a common feature in parasitic plants as a result of the shift from autotrophic to parasitic life, which is accompanied by several changes, such as pseudogenization, gene loss, structural rearrangement and size reduction [24,25]. Hemiparasites and holoparasite species of the Loranthaceae family have a genome size ranging from 116–139 kb (mean = 122.3 kbp), and a reduction in the plastid of some holoparasites may be more than that, such as in root parasites species of Cynomoriaceae the total Plastid genome length is 45,519 bp [26].

The organization, size, and structure of the chloroplast genome in *Plicosepalus* spp. are analogous to those of other angiosperms, with a typical four-region structure in the chloroplast genome. The size of the LSC regions is 69,497 bp and 69,947 bp, that of the SSC is 6038 bp and 6187 bp, and that of the two IR regions is 22,323 bp and 22,476. The LSC region in angiosperms ranges from 80–90 kb, the SSC regions are approximately 16–27 kb, and the two IRs range from 20–28 kb [27]. Hence, the reduction in the LSC and SSC regions was greater than that in the IR region in the *P*. *acaciae* and *P*. *curviflorus* plastomes.

*P*. *acaciae* and *P*. *curviflorus* had 106 and 108 genes, including 63 protein-coding genes and 25 tRNA and 4 rRNA genes for each species. A typical angiosperm chloroplast genome consists of 113 genes, including 79 protein-coding genes, 30 tRNA genes and four rRNA genes [21]. The cp genome in autotrophic *E*. *scandens* (Santalales) consists of almost the same gene number as that observed in angiosperms at 112 genes, including 79 protein-coding genes, 29 tRNA genes and 4 rRNA genes [26]. Other Santalales hemiparasites genomes have gene totals ranging from 80 to 101 [1,2,26,27,28]. Compared to other parasites in Santalales, fewer genes were lost in the *P*. *acaciae* and *P*. *curviflora* plastid genomes. The level of degradation in the chloroplast genome of parasitic plants varies with the level of photosynthesis [1]. Gene losses in the chloroplast genome of hemiparasitic Santalales are not as large as those in holoparasitic plastid genomes, which lose most or all of their genes [1]. *P*. *acaciae* and *P*. *curviflora curviflorus* have well-configured green leaves.

We also found that the AT content was higher than the GC content in the *P*. *acaciae* and *P*. *curviflorus* cp genomes, which is also observed in the chloroplast genome of holoparasitic species of the Balanophoraceae family [28,29]; nonetheless, some Angiosperms also have high AT contents [30].

The two genes *clpP* and *ycf3* had two introns in each of the *P*. *acaciae* and *P*. *curviflorus* plastomes, while the remaining genes presented only one intron, which is consistent with previous reports in the chloroplast genome *Macrosolen* (Santalales) [31] and other cp genomes of angiosperms [32,33].

Codons encoding the amino acid leucine were the most frequent while those encoding cysteine and tryptophan were the least frequent in *P*. *acaciae* and *P*. *curviflorus*, respectively. Similar results were reported in the chloroplast genomes of the order Santalales [31], as well in some species of angiosperms [32,33]. The results obtained from the present study show that in the plastid genome of *P*. *acaciae* and *P*. *curviflorus*, the amino acid exchange of serine-to-leucine represents the greatest codon transformation. The authors of [31] indicated that the amino acid conversion from serine (S) to leucine (L) occurred most frequently in three parasitic species of the *Macrosolen* genus (Santalales). This agreement between results could be attributed to the high preservation of RNA editing [34,35].

A total of 164 and 155 SSRs are present in the plastid genome of *P*. *acacia* and *P*. *curviflorus*, and most of the microsatellites in repeats are present in the noncoding IGS region. Several reports [36,37,38,39,40] have shown the importance of chloroplast SSRs (cpSSRs) as reliable molecular markers to discriminate between specimens at lower taxonomic levels and for studying the population structure. A previous study using RAPD markers revealed the difference in genetic makeup depending on the host on which the mistletoe grows. The difference in genetic makeup might influence the chemical composition and, in turn, might affect the therapeutic properties of the mistletoe [41]. Thus, we recommend developing the SSRs (cpSSRs) from the cp genome of *Plicosepalus* spp. or available hemiparasitic plastomes to investigate changes in the genetic structure of the parasitic species when the grow on different hosts and to account for variations within the population of mistletoe.

A heatmap of genes present in the plastomes of *P*. *acaciae* and *P*. *curviflorus* and the comparison species showed that all *ndh* genes (A, B, C, D, E, F, G, H, I, D and K) were lost or pseudogenes. The gene *ndh* represents a complex group consisting of approximately 30 subunits, with 11 out of 30 used for encoding subunits of the NADH dehydrogenase complex in plant plastids and involved in photosynthesis [42]. The partial or complete loss (physically or functionally) of genes associated with photosynthesis (*ndh*) is a common phenomenon in hemiparasitic and holoparasites of Santalales and Orobanchaceae [1,2,20,43,44,45].

In the present study, the *infA* gene was lost in *P*. *acaciae* and *P*. *curviflorus* and in most of the compared parasitic species, or it was present as a pseudogene. The *infA* gene is thought to function as a translation initiation factor that assists in the assembly of the translation initiation complex [21]. The *infA* gene is also a common gene lost in parasitic species of Santalales; however, the *infA* gene was also lost in the cp genome in some autotrophic species of angiosperms [46]. The *infA* gene is believed to be the most mobile chloroplast gene and is transferred to the nucleus in angiosperms [44], and it is believed that a similar scenario could also occur in parasitic plants. Several hypotheses have been proposed to explain the reasons for cp gene transfer to the nucleus by [45,46,47,48,49] and was summarised by [49] as three factors: (1) the relatively high frequency of organellar DNA escape to the nucleus provides numerous opportunities for successful functional gene transfers and is essentially a one-way process; (2) the progressive accumulation of detrimental mutations in asexual organelle genomes by Muller’s ratchet favours transfer; and (3) smaller, streamlined organelle genomes are favoured selectively. Although all of these hypotheses could be applicable to eukaryote organelles, the first hypothesis is likely more significant in flowering plants [44]. In the history of chloroplast evolution, the *infA* gene has rapidly moved from the chloroplast genome to the nuclear genome. No other chloroplast genes have been reported to undergo multiple evolutionary transfers to the nucleus. Several hypotheses have been proposed to explain why genes are transferred more than once from organelles to plastids, but many have not been applied to *infA* gene [44]. Alternatively, it could be that the small size of *infA* plays a role in repeated transfer of a gene to the nucleus, because it affects both the possibility of transferring genes to the nucleus and the prospect of the transferred gene being damaged by mutation [44].

Additionally, two rRNA (rpI32, rps16) genes and four tRNA genes (*trnG-UCC*, *trnI-GAU*, *trnK-UUU*, *trnV-UAC*) were lost in the two cp genomes of *P*. *acaciae* and *P*. *curviflorus*, and these genes are missing from most of the parasitic plastomes included for comparison. Previous studies indicated that these genes were lost in some species of the family Loranthaceae [4]. However, this loss was not limited to parasitic plants only because similar cases of gene loss were reported in angiosperms [1].

In contrast, previous studies [4,20] indicated the loss or pseudogenesis of some protocode genes, including *rp16*, *rps15*, *ycf1*, *ycf15* and tRNA (*trnA-UGC*, *trnG-GCC*, *trnH-GUG*, *trnI-CAU*, *trnI-UAA*, *trnP-UGG* and *trnL-UAG*). However, we noticed the presence of these genes in the *P*. *acaciae* and *P*. *curviflorus* cp genomes. Our results are consistent with that of [4], who found some of these missing genes after a cp genome reannotation (*ycf1*, *trnH-GUG*, *trnL-UAG* and *trnL-UAA*).

Overall, comparisons between the plastomes of *Plicosepalus* spp. and hemiparasite species showed different degrees of gene loss. A similar pattern of gene loss from the plastome was reported in parasitic species of the family Orobanchaceae as well [20]. Hemiparasites still present varying degrees of photosynthetic ability and use their host to meet some of their nutrient needs, whereas holoparasites lose their photosynthetic ability and depend entirely on the host plant to meet their nutrient needs. Consequently, hemiparasitic plants have different degrees of photosynthesis ability; thus, the chloroplast genomes of parasitic plants are exposed to different levels of selective pressures [24,46]. We suggest that further studies should be conducted on the Plastome structure of different types of hemiparasitic plants to determine the exact structural changes that occur in the genome and the variation in genes lost during the shift to the parasitic state.

Although close species tend to have similar IR/SC boundaries, several studies have reported variations in the size and boundaries among IR/LSC and IR/SSC regions and variations in the gene location [47,48]. We observed variations in the size of the IR region in the hemiparasitic species. In addition to a reduction in the size of the LSC region and a large reduction in the size of the SSC, a greater amount (more than half) of the SSC region size was lost compared to that in angiosperms [27], which could be the result of contraction and expansion of the IR region.

The *rpl2* gene occupied the LSC/IRb borders in all samples, which is consistent with previous observations [31] in parasitic species of Santalales. In addition, *ycf1* was found at the IR/SC borders in most of the compared species, which is consistent with the results for Loranthaceae species mentioned in previous studies [14,31,50,51]. The IRa/SSC boundaries of the two species *P*. *acaciae* and *P*. *curviflorus* were distinguished by the presence of the protein-code gene *rps15*, which is usually lost from many other hemiparasites species, as mentioned by [4,20].

Zong et al. indicated that the *trnL* gene is present in the SSC in the *Macrosolen* genus of the Loranthaceae family [30]. We noticed the same phenomenon in certain species, and the *trnL* gene was extended along the SSC/IRb border in certain species, which could be attributed to the extension in the IR region. The *ycf1* gene present at the SSC/IRb borders of angiosperm plastomes is often pseudogenized, and the expansion length of *ycf1* could influence the IR length and the gene distribution at the SC/IR borders [52,53,54].

The *trnH* gene is usually located at the IRb/LSC border in angiosperms. We observed variations in the genomes of *Plicosepalus* spp. and the comparison species, where the *trnH* gene was present in the inverted repeat region, which could be a result of expansion in the IR region. A similar finding was reported by [55] in the Acanthoideae family.

The values of synonymous (Ks) and nonsynonymous (Ka) substitutions and the Ka/Ks ratio showed that protein-coding genes (*petB*, *psbM*, *accD*, *ycf1*, *rpl23* and *atpI*) were under positive selection in the *P*. *acaciae* and *P*. *curviflorus* chloroplast genomes, and these genes could have a faster evolution rate [56]. Our result corresponds with that of [31], who reported that the ycf1 gene was a mutational hotspot in hemiparasitic *Macrosolen* species (Santalales). We suggest subjecting these genes to further investigation to identify their ability as indicators of phylogenetic relationships within Loranthaceae.

The plastome consists of many highly efficient genes capable of resolving phylogenetic issues at different levels of angiosperm taxonomy [33,57,58,59]. In this study, we found that *P*. *acaciae* and *P*. *curviflorus* were strongly related to the family Loranthaceae and tribe Lorantheae and all species in the family Loranthaceae represented the monophylitic group. These findings provide additional evidence to confirm the monophyly state of Loranthaceae, as reported by previous studies [3]. In addition, they confirm the taxonomic state of root hemiparasitic *N*. *floribunda* (tribe Nuytsieae) and *E*. *albida* (tribe Elytrantheae) as sisters to the Lorantheae tribe (belonging to the family Loranthaceae), as mentioned by [3,5,60].

The current results supported those of previous studies [3]. In the taxonomic state of genera and species belonging to subtribes of Lorantheae, the results corresponded to that of [3], who reported that the subtribe Dendrophile did not represent a monophyletic group. In contrast, we noticed a difference in the taxonomic case of *M*. *cochinchinensis*, which was previously reported to belong to the tribe Elytrantheae [3]; however, it is a part of the Lorantheae tribe in the current ML and MP trees in a highly supported clade (BP = 100), suggesting the need to rearrange some genera and change the circumscription of some tribes and subtribes of Lorantheae.

## 4. Materials and Methods

### 4.1. Sample Collection

Fresh leaves of *P*. *curviflorus* were collected from the Al-Taif region, Saudi Arabia (21.218874°, 40.557426°), while those of *P*. *acacia* were collected from the Al-Wajh district, Saudi Arabia (26.5737260°, 36.3731150°). The plants were identified by Dr. Rahma Alqthanin, the curator of the Sultan bin Abdulaziz for Research and Environmental Studies Centre, based on herbarium specimens and morphologies in the relevant literature. A sample specimen was prepared and deposited in the herbarium of Umm Al-Qura University, Makkah, under accession numbers *P*. *curviflorus* (UQU012021) and *P*. *acacia* (UQU062021). Samples of fresh leaves were dried in silica gel for DNA extraction. DNA was extracted from the silica gel-dried leaves of *P*. *curviflorus* and *P*. *acacia* using the CTAB Plant DNA extraction protocol [61].

### 4.2. Library Construction

To build the genomic library, 1.0 µg g of DNA was used as input material for sample preparation. The DNA library was constructed using the Illumina TruSeq Nano DNA 350 Kit (Illumina, San Diego, CA, USA) following the manufacturer’s recommendations. The library was initially prepared via the random fragmentation of DNA samples to a size of 350 bp, followed by ligation to 5′ and 3′ adapters. The adapter-ligated fragments were then amplified via PCR and subjected to gel purification. To verify the size of the PCR-enriched fragments, the template size distribution was checked with an Agilent Technologies 2100 Bioanalyzer using a DNA 1000 chip (Agilent Technologies, Santa Clara, CA, USA). The prepared libraries were quantified using qPCR in accordance with the Illumina qPCR Quantification Protocol Guide (Illumina, San Diego, CA, USA).

### 4.3. De Novo Genome Sequencing

Library construction and sequencing was performed using Illumina sequencing (Illumina, San Diego, CA, USA) and a read length of 151 bp paired ends, and the procedures were carried out by Macrogen (https://dna.macrogen.com/, Seoul, Korea). The final yield of filtered data was 3.7 Gb and 3.25 Gb for *P*. *curviflorus* and *Plicosepalus acacia*, respectively.

### 4.4. Genome Assembly and Annotation

The FastQC tool was used to assess the raw read quality using a Phred score above 30. All adapters were removed, vector contamination was removed from the assembly, and the N50 value was high for a single genome. Clean reads were processed for genome assembly using NOVOPlasty 4.3.0 Version [62] with kmer (K-mer = 33) to assemble the complete chloroplast genome from the whole genomic sequence of *Plicosepalus* spp. *Arabidopsis thaliana* (NC_000932.1) was used as a reference in the assembly. Single contigs containing the complete chloroplast genome were generated. Gene prediction and annotation of the *Plicosepalus* spp. chloroplast (cp) genome were performed using the GeSeq tool [63] with default parameters, and the percent identity cut-off for protein-coding genes and RNAs was set at ≥60 and ≤85, respectively. tRNA genes were identified with trnAscan-SE Version 2.0 [64]. The annotated (gb) format sequence files were used to draw the circular chloroplast genome maps with the OGDRAW tool (Organellar Genome DRAW), Version 1.3.1 [65]. The sequences of the chloroplast genome of *Plicosepalus* spp. were deposited in the GenBank database under accession numbers *P*. *acacia* (OM640467) and *P*. *curviflorus* (OM675776).

### 4.5. Sequence Analysis

The relative synonymous codon usage (RSCU) values, base composition, and codon usage were analyszd using MEGA software [66], Version 11.0. Potential RNA editing sites present in the protein-coding genes were predicted by the PREP suite [21], with a cut-off value of 0.8.

### 4.6. Repeat Analysis in the Chloroplast Genome

The online software MIcroSAtellite (MISA) v2.1 [67] was used to identify simple sequence repeats (SSRs) in the chloroplast genome of *Plicosepalus* spp. and six other species from the Loranthaceae family, namely, *S*. *chingii*, *T*. *chinensis*, *L*. *europaeus*, *D*. *pentandra*, *N*. *floribunda*, and *E*. *albida*. Parameterwise, eight, five, four and three repeat units were assessed for mononucleotides, dinucleotides, trinucleotides and tetra- and pentanucleotide SSR motifs, respectively.

In addition, REPuter [68] software was used with default settings to detect the size and location of long palindromic, forward, reverse, and complementary repeats in the *Plicosepalus* spp. cp genomes and the genomes of six species from Loranthaceae.

### 4.7. Sequence Divergence and Boundary

Comparisons between the genome of *Plicosepalus* spp. and six chloroplast genomic sequences of Loranthaceae (*S*. *chingii*, *T*. *chinensis*, *L*. *europaeus*, *D*. *pentandra*, *N*. *floribunda*, *E*. *albida*) were analysed using the mVISTA program [69], and the annotation of *Plicosepalus* spp. was used as a reference in the Shuffle-LAGAN mode. Furthermore, comparisons between the borders of the IR, SSC, and LSC regions were performed using IRSCOPE [70].

### 4.8. Characterization of the Substitution Rate

Methods for estimating nonsynonymous and synonymous substitution rates (Ka and Ks), selection and beneficial mutations among protein-coding sequences were applied [71]. Nonsynonymous (Ka), synonymous (Ks), and Ka/Ks ratios were calculated to detect plastome genes under selection pressure in *Plicosepalus* spp. and they were compared with those in the six aforementioned Loranthaceae species. We employed KaKs Calculator Version 2.0 [71] with default parameters and the Nei and Gojobori substitutions.

### 4.9. Heatmap

A heatmap was created to investigate the evolution of the plastome associated with parasitism in the two plastomes of the aerial hemiparasites evaluated in this study, i.e., *P*. *acacia* and *P*. *curviflorus*, and the results were compared to that of the species of the Loranthaceae family, *S*. *chingii*, *T*. *chinensis*, *L*. *europaeus*, *D*. *pentandra*, and *E*. *albida*. *N*. *floribunda* represents a root hemiparasite in Loranthaceae. In addition to the aerial hemiparasitic species *V*. *album* (Viscaceae), *S*. *jasminodora* is a root hemiparasite (Santalaceae). *E*. *virginiana* was used in the heatmap as a representative of an obligate parasite or holoparasite parasite (Orobanchaceae). *E*. *scandens* (Santalales; Erythropalacea) was included as an example of an autotrophic plant. Heatmaps were generated using Python Version 3.7 [72], and multiple libraries were used to plot the heatmap graph. Pandas was used to import data, and Seaborn and plotly were used to plot the actual heatmap data.

### 4.10. Phylogenetic Analysis

Phylogenetic analyses were conducted built based on coding genes of genome sequences of Santalales order, including species and families (Loranthaceae, Santalaceae, Schoepfiaceae and Viscaceae). Species of the Solanaceae family (*N*. *tabacum*) were used as an outgroup. To identify gene families, the OrthoFinder (v 2.5.4) pipeline [73] was sequentially applied to the genomes with all-to-all BLASTP (E-value ≤ 1 × 10^− 5^), reciprocity best hit, pairs connected by orthology and in-paraolgy, normalize the E-value and cluster pairs by OrthoFinder. Finally, genes were classified into orthologues, paralogues and single copy orthologues (only one gene in each species). To construct the phylogenetic tree, single-copy orthologous genes were used; each gene family nucleotide sequence was aligned using Mafft [74], and the phylogenetic tree was built with both the maximum likelihood (GTR model) and the maximum parsimony using Megatool [66], supports for nodes were assessed with 1000 bootstrapping replicates. The cladograms of the two methods were compared, and we considered that the evolutionary tree constructed by the ML method was more fit. The phylogenetic tree was visualized and modified by ITOL [75].

## 5. Conclusions

The aim of the present research was to provide the complete chloroplast genome of the medicinal species and hemiparasitic species *P*. *acaciae* and *P*. *curviflorus* to assess the systematic relationships within the Loranthaceae family. Furthermore, to investigate the evolution of plastome structure associated with parasitism, we compared these the cp genomes of these species with examples of available hemiparasitic and holoparasitic species and autotroph plants. We observed a reduction in the genome size of hemiparasitic *P*. *acaciae* and *P*. *curviflorus* and a loss of some genes. However, these losses were much less than those observed in the hemiparasite and holoparasite cp genomes. For a better understanding, we recommend that future studies investigate the chloroplast genome of different species in families of Santalales. This study confirmed the taxonomic status of the species *Plicosepalus acaciae* and *P*. *curviflorus* as members of the Loranthaceae family and Lorantheae tribe; however, some species of the Loranthaceae family still require further investigation of their taxonomic status. Moreover, available genome data could facilitate more advanced applications in parasitic plant research, contribute to good parasitic weed management and contribute to modifying the host species to become more resistant.

## Figures and Tables

**Figure 1 plants-11-01869-f001:**
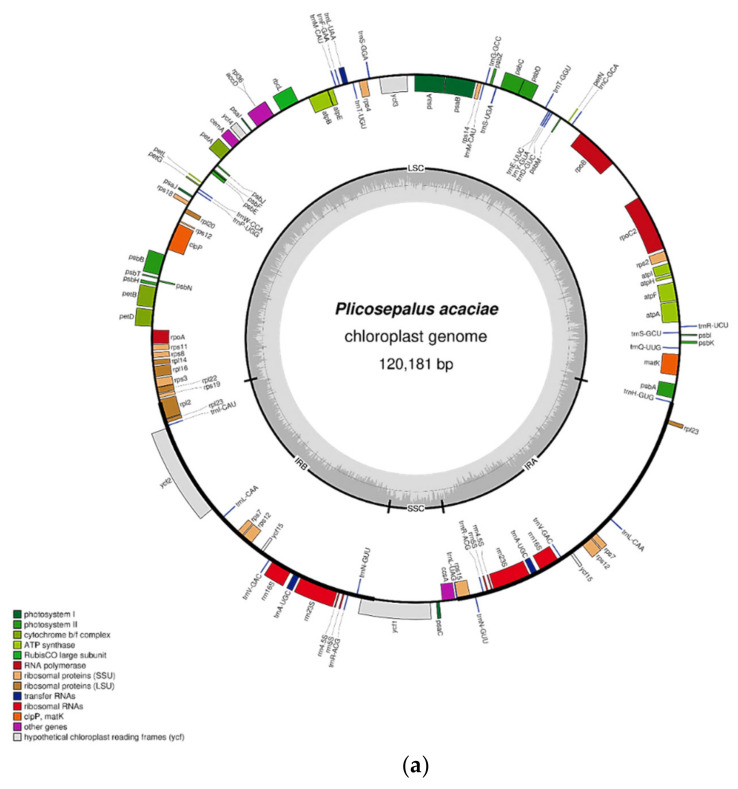
(**a**) Gene map of the *Plicosepalus acaciae* plastid genome. (**b**) Gene map of the *Plicosepalus curviflorus* plastid genome. Small single copy (SSC), large single copy (LSC) and inverted repeats (IRa and IRb) are indicated. Thick lines on the outer complete circle identify the inverted repeat regions (IRa and IRb). The genes outside the circle are transcribed counterclockwise, whereas those inside the circle are transcribed clockwise. Genes belonging to different functional groups are highlighted in different colours. The dark grey area in the inner circle indicates the CG content of the plastome, using OGDRAW tool Version 1.3.1.

**Figure 2 plants-11-01869-f002:**
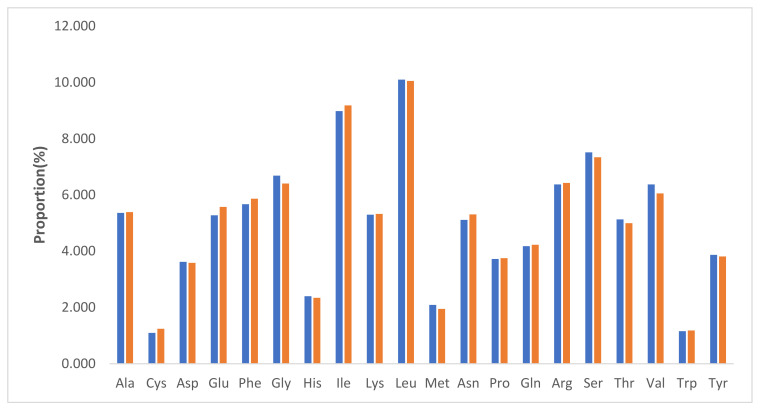
Amino acid frequencies of the protein-coding sequences of *Plicosepalus acaciae* (blue) and *Plicosepalus curviflorus* (orange) chloroplast genomes using MEGA software Version 11.0; the most and least frequent amino acids are shown.

**Figure 3 plants-11-01869-f003:**
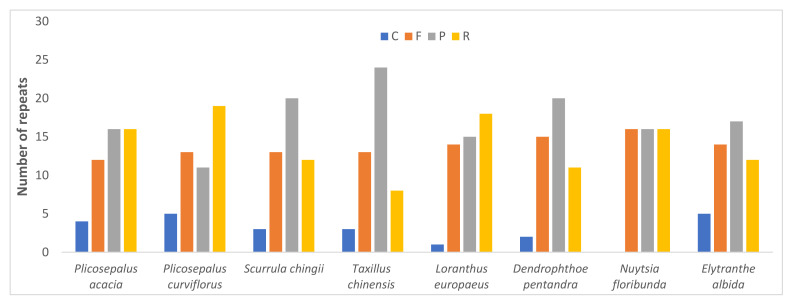
Number of each repeat type—F, forward; P, palindromic; R, reverse; and C, complement repeats—in the plastid genome of *Plicosepalus acaciae* and *Plicosepalus curviflorus* and six species from Loranthaceae, using REPuter 2 software. *Taxillus chinensis* had the highest frequency of palindromic repeats, *Nuytsia floribunda* had the highest frequency of forward repeats, and *Plicosepalus curviflorus* had the highest frequency of reverse repeats. Complement repeats were the least common type of repeat.

**Figure 4 plants-11-01869-f004:**
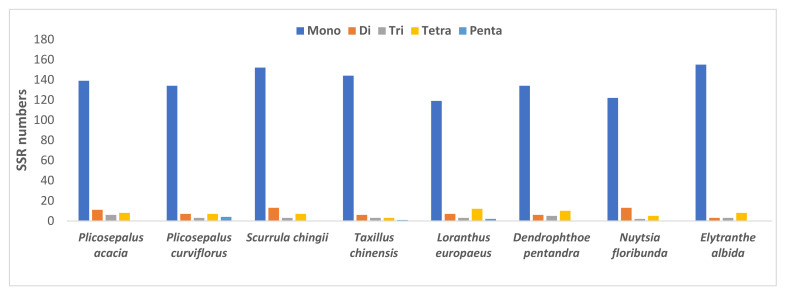
Number of different simple sequence repeat (SSR) types in the plastid genomes of *Plicosepalus acaciae* and *Plicosepalus curviflorus* and six species from Loranthaceae using MISA software v2.1. The majority of SSRs in the cp genome were monorepeats.

**Figure 5 plants-11-01869-f005:**
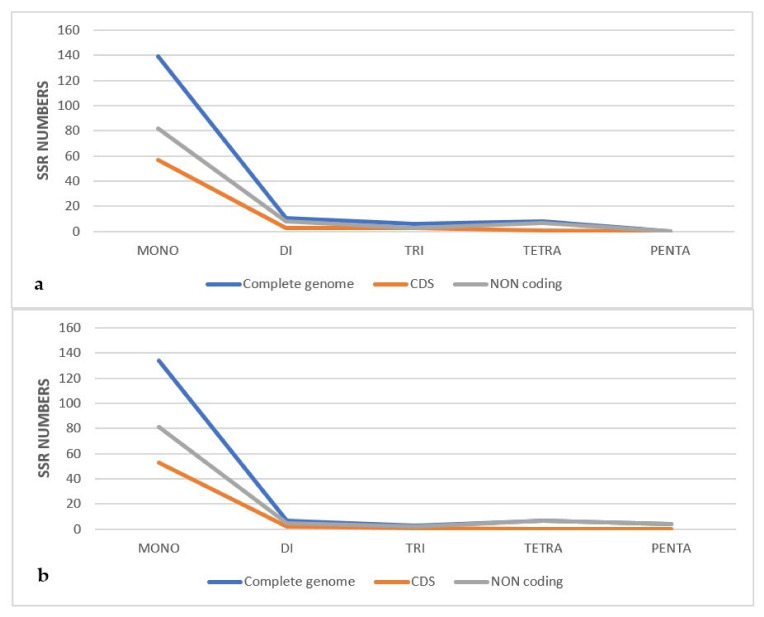
Number of SSR types in the complete chloroplast genome, protein-coding regions, and non-coding regions of (**a**) *Plicosepalus acacia*, (**b**) *Plicosepalus curviflorus*.

**Figure 6 plants-11-01869-f006:**
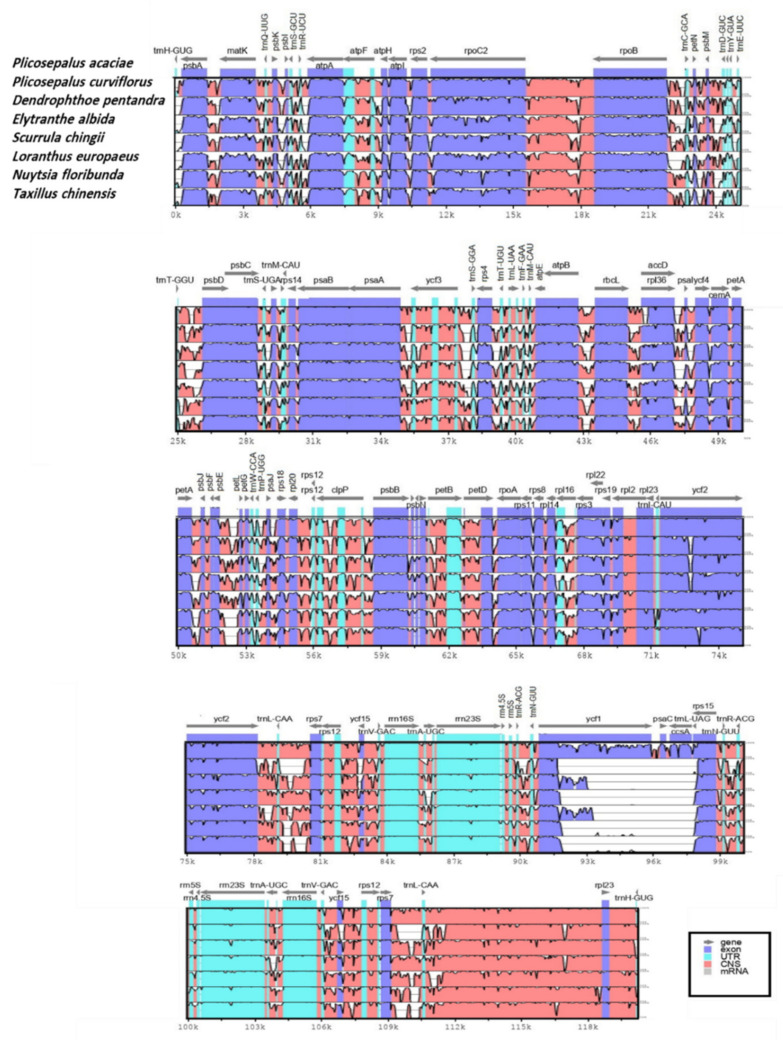
Whole chloroplast genome alignments for Loranthaceae species via the mVISTA program, using the annotation of *Plicosepalus acaciae* as reference. The *x*-axis represents the coordinates in the cp genome, while the *y*-axis indicates percentage identity from 50% to 100%. The top grey arrows indicate the position and direction of each gene. Pink indicates non-coding sequences (NCS), blue indicates protein-coding genes, and light green indicates tRNAs and rRNAs.

**Figure 7 plants-11-01869-f007:**
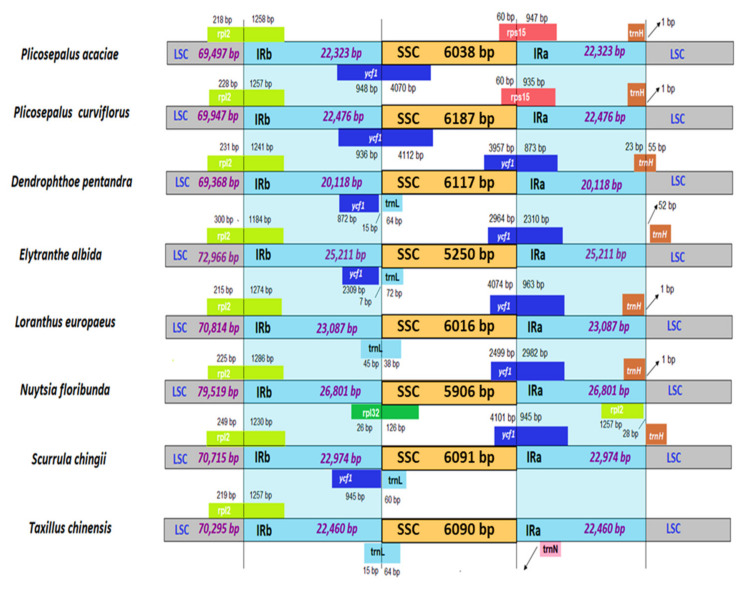
Comparison of the large single copy (LSC), a small single copy (SSC) and two inverted repeats (IRa and IRb) region borders among the chloroplast genomes of eight Loranthaceae species using IRSCOPE. Variations in the region’s length and gene locations are observed.

**Figure 8 plants-11-01869-f008:**
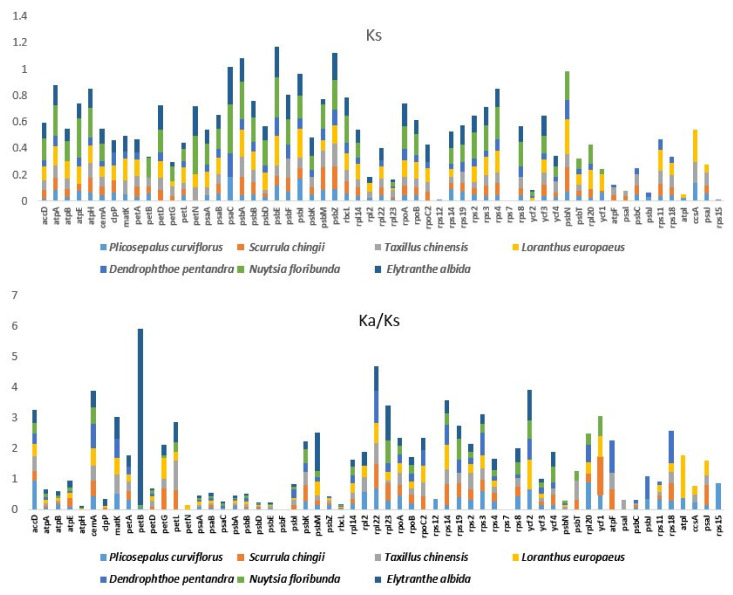
Synonymous (Ks) and Ka/Ks ratio values of 59 protein-coding genes of the *Plicosepalus acacia* vs. *Loranthaceae plastomes* (*Plicosepalus curviflorus*, *Scurrula chingii*, *Taxillus chinensis*, *Loranthus europaeus*, *Dendrophthoe pentandra*, *Nuytsia floribunda* and *Elytranthe*), using the KaKs Calculator 2.0 to detect substitution, selection, and beneficial mutation genes under selective pressure (>1).

**Figure 9 plants-11-01869-f009:**
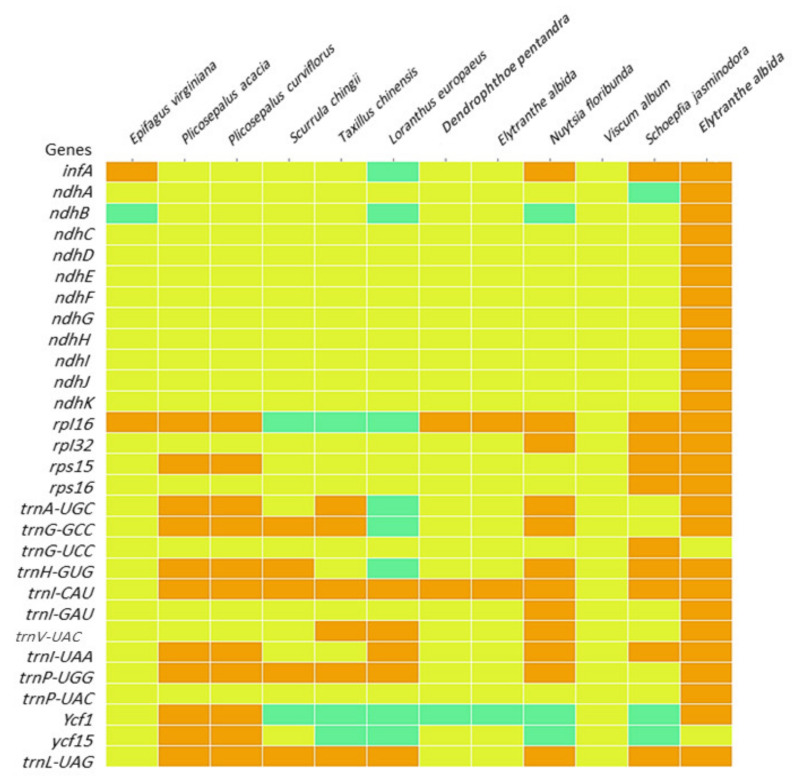
Heatmap displaying a comparison of the plastid genome gene content of 11 parasitic plants and 1 autotrophic plant (*Erythropalum scandens*) using Plotly software. The common existing genes in the plastid genome of the 12 species are not listed. Orange colour indicates each gene present and seafoam indicates a pseudogene. The yellow indicates an absent gene.

**Figure 10 plants-11-01869-f010:**
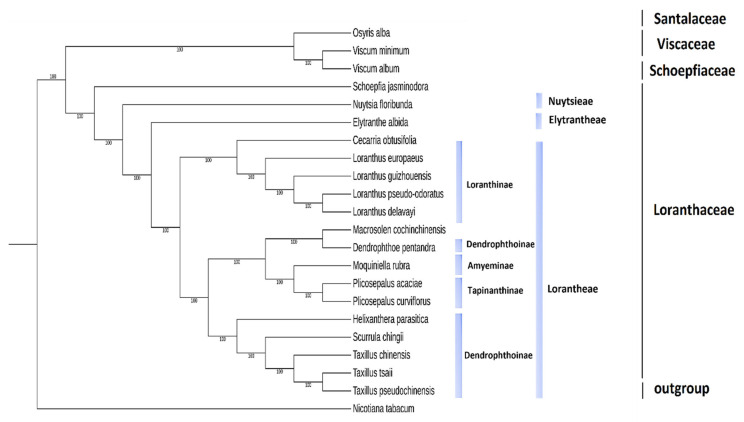
Phylogenetic tree construction inferred from the complete chloroplast genomes of 21 taxa including *Plicosepalus acaciae* and *Plicosepalus curviflorus*, using Maximum Likelihood (ML) methods and Megatool. The tree shows the relationships between tribes and sub-tribes of Loranthaceae and related families of the order Santalales (Viscaceae, Santalaceae, and Schoepfiaceae), *Nicotiana tabacum* was used as an outgroup. The numbers in the branch nodes represent bootstrap support (BS). All branches of the tree were highly supported with 100% bootstrap values. Monophyly of the Loranthaceae family was strongly supported and *P*. *acacia* and *P*. *curviflorus* belong to subtribe Tapinanthinae.

**Table 1 plants-11-01869-t001:** Chloroplast genome features of *Plicosepalus acaciae* & *Plicosepalus curviflorus*.

Feature	*P*. *acaciae*	*P*. *curviflorus*
Genome size (bp)	120,181	121,086
IRA (bp)	22,323	22,476
IRB (bp)	22,323	22,476
LSC (bp)	69,497	69,947
SSC (bp)	6038	6187
Total No of Genes	106	108
Total No of Unique Genes	92	92
rRNA	4	4
tRNA	25	25
Protein-Coding genes	63	63
A%	31.50	31.34
T (U) %	31.93	31.89
G%	17.97	18.12
C %	18.59	18.65
%GC	36.6	36.8

**Table 2 plants-11-01869-t002:** Genes with introns in the chloroplast genome of *Plicosepalus acaciae*.

Gene	Strand	Location	Exon I (bp)	Intron I (bp)	Exon II (bp)	Intron II (bp)	Exon III (bp)
atpF	-	LSC	145	801	401		
ycf3	-	LSC	124	762	230	768	153
clpP	-	LSC	71	767	294	675	229
petB	+	LSC	6	814	642		
rps12	-	IRB	56	515	229		
	+	IRA	229	515	56		
rpl2	-	IRB	391	654	431		
rpl16	-	LSC	303	486	9		
petD	+	LSC	8	743	484		
trnL-UAA	+	LSC	35	299	55		
trnA-UGC	-	IRA	55	342	39		
	+	IRB	39	342	55		

**Table 3 plants-11-01869-t003:** Genes with introns in the chloroplast genome of *Plicosepalus curviflorus*.

Gene	Strand	Location	Exon I (bp)	Intron I (bp)	Exon II (bp)	Intron II (bp)	Exon III (bp)
atpF	-	LSC	145	792	410		
rpoC1	-	LSC	432	830	1617		
ycf3	-	LSC	124	743	230	772	153
clpP	-	LSC	71	752	294	524	229
rpl2	-	IRB	391	654	440		
rps12	-	IRB	56	507	229		
	+	IRA	229	507	56		
rpl16	-	LSC	9	893	81		
petB	+	LSC	6	864	642		
trnL-UAA	+	LSC	35	290	55		
trnA-UGC	-	IRA	55	350	37		
	+	IRB	37	350	55		

**Table 4 plants-11-01869-t004:** cpSSRs detected in eight Lornthaceae chloroplast genomes.

SSR Type	Repeat Unit	*Plicosepalus* *acacia*	*Plicosepalus curviflorus*	*Scurrula* *chingii*	*Taxillus* *chinensis*	*Loranthus* *europaeus*	*Dendrophthoe pentandra*	*Nuytsia floribunda*	*Elytranthe* *albida*
Mono	A	67	62	59	49	44	50	47	60
	C	2	2	4	7	6	6	3	6
	G	1	0	2	2	2	0	3	2
	T	69	70	87	86	67	78	69	87
Di	AG/CT	1	1	0	0	0	1	1	0
	AT/AT	10	6	13	6	7	5	12	3
Tri	AAG/CTT	2	2	2	2	2	2	0	0
	AAT/ATT	4	1	1	1	1	3	2	1
	ATC/ATG	0	0	0	0	0	0	0	2
Tetra	AAAG/CTTT	3	3	1	1	2	2	0	3
	AAAT/ATTT	4	2	1	0	4	5	3	1
	ACAG/CTGT	1	0	1	1	1	1	0	1
	AGGG/CCCT	0	2	2	0	2	2	0	0
	AAGT/ACTT	0	0	1	0	0	0	0	0
	AATT/AATT	0	0	1	0	0	0	0	1
	AATC/ATTG	0	0	0	1	0	0	0	0
	AAAC/GTTT	0	0	0	0	1	0	0	0
	AACC/GGTT	0	0	0	0	1	0	0	0
	AGAT/ATCT	0	0	0	0	1	0	1	1
	AATG/ATTC	0	0	0	0	0	0	1	1
Penta	AAAAT/ATTTT	0	1	0	0	0	0	0	0
	AATAT/ATATT	0	3	0	1	2	0	0	0
Total		164	155	175	157	143	155	142	169

**Table 5 plants-11-01869-t005:** Accession numbers of chloroplast genome analyzed in the study.

Order	Family	Accession Number	Organism
Santalales	Loranthaceae	NC_053563	*Scurrula chingii* (W.C.Cheng) H.S.Kiu
		MN080717	*Taxillus chinensis* (DC.) Danser
		MN080718.1	*Helixanthera parasitica* Lour.
		MT987630	*Loranthus europaeus* Jacq.
		MT987635.1	*Loranthus pseudo-odoratus* Lingelsh.
		NC_039376	*Macrosolen cochinchinensis* (Lour.) Tiegh.
		NC_045107	*Dendrophthoe pentandra* (L.) Miq.
		NC_045108	*Elytranthe albida* (Blume) Blume
		NC_058837	*Taxillus pseudochinensis* (Yamam.) Danser
		NC_058840	*Taxillus tsaii* S.T.Chiu
		NC_058859	*Cecarria obtusifolia* (Merr.) Barlow
		NC_058862.1	*Loranthus guizhouensis* H.S.Kiu
		NC_058868	*Moquiniella rubra* (A.Spreng.) Balle
		NC_058869	*Nuytsia floribunda* (Labill.) R.Br. ex G.Don
		NC_058858.1	*Loranthus delavayi* Tiegh.
	Santalaceae	NC_027960	*Osyris alba* L.
	Schoepfiaceae	NC_034228	*Schoepfia jasminodora* Siebold & Zucc.
	Viscaceae	KT003925.1	*Viscum album* L.
		NC_027829	*Viscum minimum* Harv.
	Erythropalaceae	NC_036759.1	*Erythropalum scandens* Blume
Lamiales	Orobanchaceae	NC_001568.1	*Epifagus virginiana* (L.) Barton
Solanales	Solanaceae	NC_001879	*Nicotiana tabacum* L.

## Data Availability

The data presented in this study are available in this article and Appendix A. The complete chloroplast genome sequence of *P*. *acacia* and *P*. *curviflorus* were deposited in GenBank at https://www.ncbi.nlm.nih.gov, (accessed on 2 February 2022) (accession numbers: *P*. *acacia* (OM640467) and *P*. *curviflorus* (OM675776).

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
