# Peer review of "Gene Losses and Plastome Degradation in the Hemiparasitic Species Plicosepalus acaciae and Plicosepalus curviflorus: Comparative Analyses and Phylogenetic Relationships among Santalales Members"

_plants, 2022, doi:10.3390/plants11141869_

Round 1
Reviewer 1 Report
The article "Gene Losses and Plastome Degradation in the Hemiparasitic Species Plicosepalus acaciae and Plicosepalus curviflorus: Comparative Analyzes and Phylogenetic Relationships among Santalales Members" by Widad AL-Juhani, Noha T. Al Thagafi, Rahmah N. Al-Qthanin is an interesting well-structured study that allows evaluate preliminary data concerning the issue of changes in the composition of some of the genes integrated into the genome of organelle. Unfortunately, the article requires significant modification, correction of figures, proofreading of the text, as it can mislead readers.
I think that this work can be done purely technically and will not take much time.
The names of plants are written with the indication of the author of the description.
The fundamental mistake is that the authors: call non-photosynthetic plastids chloroplasts. This is completely false and obvious. As if the authors have very little knowledge of the biology of the plant cell. The terminology throughout the text and in the captions should be corrected. Obviously, there is no specific chloroplast genome, but there is a plastid genome.
The absence of a discussion of the role of plastid genes, which are introduced into the nuclear genome as a result of evolutionary transformations, leaves an extremely strange impression, while it is obvious that their role is no less important. It is clear that the authors did not attach importance to this issue. However, it is strange not to discuss these data in terms of the relationship with the nuclear genome. In any case, this needs to be mentioned. Also, the description of chloroplasts in those parts of the manuscript that deal with plastids should be excluded. It is equally important to correct the text of the introduction and discussion and add a description and sources indicating the correct names of non-photosynthetic plastids that the authors can presumably be talking about. The range of such plastids is quite extensive and includes etioplasts, chromoplasts, amyloplasts, etc. These plastids are so different from each other that they could be mistaken for completely different organelles. However, they have the same genome, and the issue lies in changing the expression patterns of specific genes. I could not find a discussion of this issue in connection with the given data.
Minor remarks:
Picture 1 is unreadable, there are no explanations for it, it should be noted that the placement of information in the form of abbreviations (even if it concerns commonly used names of genes, sites, or restrictases is mandatory in all cases) there are special methods for this, a list of abbreviations, etc.
Figure 3 contains abbreviated names of plants; there are no full names and explanations in the figure caption. It should be borne in mind that in modern scientific literature the drawing is considered as an independent result and cannot be obscure. The method, object, and result must be obvious from the combination of the image and the caption.
The same remarks apply to Figure 4.
In Figure 5, the signature contains the cryptic term "non-coding genes" should explain what is meant or correct the text.
Figure 6 is almost unreadable, and if it cannot be corrected and it does not make much sense, put it in the Supplementary Materials. Another option is to make the signatures and numbers visible. Also, I was unable to detect mRNA, although it is obvious that it is there. Is it possible to just stretch the image?
Figure 7 Again, there is no interpretation of LSC, SSC, and IR ; but there is a mysterious chloroplast genome, the same problem with species names.
In Figure 8, there is a problem with plant names, gene designations, units of measurement on the scale.
Figure 9. Still the same problems + nothing is visible, and the color names do not match the image, by light green they mean yellow, and dark green is the color of the sea wave? The chloroplast genome is here again, not the plastid one.
Figure 10 is an incomplete title and the absence of an indication of what the numbers 100 and 99 mean. I wonder why tobacco is deprived of a number and does it make sense?
The materials and methods do not indicate the firms that produce the programs.
Due to the fact that I consider the manuscript interesting, I consider it possible to consider it in the major revision mode after a significant revision of the entire text.
Reviewer 2 Report
This manuscript is devoted to a very interesting comparative study of complete chloroplast genomes of hemiparasitic species. This work meets the requirements for such studies. However, there are a number of minor comments.
Line 80 It is better to specify “Liu et al. [2] conducted…”
Line 254 Table 4 shows the results of the microsatellite frequency of not 6 but 8 species. Or it should be “P. acaciae and P. curviflorus with the six…”
Since you are comparing eight species, count the polymorphic SSR loci from the presented loci
Line 238 The Latin name (T. chinensis) is briefly and then again full. Uniformity is needed. List all Latin names that occur repeatedly, all in short (usually) or full form.
Figure 8 Why is the Ka/Ks ratio not investigated for Plicosepalus acacia?
Line 507 Plicosepalus acaciae and Plicosepalus curviflorus are listed in full form and were previously in short form.
Line 514 and 515 should not be written from the red line.
Line 544-552 Indicate whether the reduction in the genomes of P. acaciae and P. curviflorus is mainly due to the genic or non-genic part.
Line 665 It is better to specify “Zong et al. [32] indicated…”
Line 705 Check it. “1.0 g of DNA was used”
Reviewer 3 Report
In this study, plastomes of two hemiparasitic species of Plicosepalus, P. acaciae and P. curviflorus, were sequenced and bioinformatic analyses were performed. The results are indeed interesting for people studying the evolution of the parasitic species of Santalales. However, this manuscript is solely bioinformatic and might fit other journals which focus on taxonomy and/or evolutionary biology. Nowadays the sequencing of plastomes is not a new technology, and this manuscript using common methods does not provide sufficient novel insights into a biological issue.
Round 2
Reviewer 1 Report
The manuscript "Gene Losses and Plastome Degradation in the Hemiparasitic Species Plicosepalus acaciae and Plicosepalus curviflorus: Comparative Analyzes and Phylogenetic Relationships among Santalales Members" has been modified and corrected by the authors. Although the work has a narrow focus, it contains elements of novelty and may be of interest to people working in the field of bioinformatics. The shortcomings of the work have been partially eliminated and the work can be accepted provided that the keywords "genome structure" are replaced by "plastom structure" and the quality of figures that remain poorly readable and have poor resolution is improved.
